# Evaluation of rural comprehensive development level and obstacle factors in various countries around the world

**Yiyong Chen, Ling Zhu, Jinzhao Du, Wuyang Hong**\*

College of Architecture and Urban Planning, Shenzhen University, State Key Laboratory of Subtropical Building and Urban Science, Shenzhen, China,

\* meszuwu@szu.edu.cn

## Abstract

At present, major developing countries in the world are entering a stage of rapid urbanization, and rural transformation and development are accelerating. Conducting international comparisons of rural development, evaluating and diagnose rural development levels of various countries and summarizing the advantages of rural areas in developed countries. These summaries and comparisons are important for developing countries to implement rural revitalization and promote the transformation of rural regional systems. Basing on the connotation of rural development, this study constructs an evaluation index system for rural development level worldwide. Through AHP-entropy method, TOPSIS method, obstacle model, and regression model, this study analyzes the rural comprehensive development level of 175 countries and regions and obstacle factors on the basis of statistical data from the United Nations and World Bank. Results suggest significant differences in the comprehensive development level of rural areas among countries at the global level. In addition. The comprehensive development level of rural areas exhibits definite correlations with national income level, per capita GDP, and the Human Development Index (HDI), among which it shows the weakest correlation with the level of urbanization. The main obstacle factors to rural development in various countries are agricultural mechanization level, agricultural land output rate, average per labor of cultivated land area, number of doctors per thousand, and per capita income of farmers. The factors affecting the rural development level are resource endowment, economic development level, education level, and government support for agriculture. Based on the research findings, this study proposes strategic recommendations for improving the rural comprehensive development level from the perspectives of agricultural and rural development and the living standards of farmers.

## Introduction

In recent years, numerous major developing countries across the globe have entered a period of rapid urbanization. With rural populations continuing to migrate to cities, rural transformation and development are accelerating. Urbanization has attracted much attention

**Data availability statement:** Data cannot be shared publicly because of the data comes from a third party, we are not authorized to share the data, Data are available from the Worldbank and FAO (contact via https://data.worldbank.org.cn/ and https://www.fao.org/faostat/zh/#data) for researchers who meet the criteria for access to confidential data. The data underlying the results presented in the study are available from https://data.worldbank.org.cn/ and https://www.fao.org/faostat/zh/#data. The sources of all data can be viewed in Table 1.

**Funding:** Fund; Major Program of the National Natural Science Foundation of China (42293273).

**Competing interests:** The authors have declared that no competing interests exist.

worldwide. Most countries are consistently encouraging urban expansion to stimulate economic development and enhance the overall quality of national life. However, the issue on rural decline, stemming from the process of urbanization, had been inadequately addressed [1,2]. Particularly in the process of urbanization in developing countries, the discord and division between urban and rural areas are severe, and the kinetic energy of rural development is insufficient. And it is accompanied by practical problems such as rural depopulation, soil erosion, and aging population, which exacerbate the risks and crises associated with rural transformation and development [3,4]. Achieving the Sustainable Development Goals (SDGs) will be difficult in a world characterized by a huge urban–rural development gap [5]. Carrying out international comparisons of rural development, evaluating and diagnosing the rural development levels of different countries, summarizing the strengths of developed nations in rural development, and identifying obstacles to rural development are crucial steps in implementing rural revitalization and promoting the transformation of rural regional systems in developing countries.

Numerous studies have been conducted on rural transformation and development, including sustainable development [6–8], development and transformation [8–10], rural revitalization [1,11–13], and rural development evaluation [14–16], among others. Quantitative evaluation of rural development level is a key concern for scholars. The earliest quantitative research on rural areas in Western countries can be traced back to the 1960s. Božović and Đurašković emphasized that more attention should be paid to human factors in future agricultural development [17], Li et al. [18] believed that achieving a balanced development of modern agriculture necessitates considering multiple dimensions, including the economy, society, population, environment, and resources. These perspectives provide valuable insights for rural development and construction in developing countries.

In terms of evaluating rural development level, many scholars have explored methods for measuring the levels of rural revitalization, urban–rural integration, or rural development. Cloke and Edwards [19] proposed an analytical framework for rural development from the perspectives of population, transportation, and land use. Gulumser et al. [20] conducted a comprehensive evaluation on rural structures in Turkey by using indicators from the Organization for Economic Co-operation and Development and European Union. Wu [21] and Zhang [22] performed evaluations of rural revitalization and rural modernization development in specific provinces, respectively. Geng et al. [23] constructed a comprehensive index system within a 5E framework for forecasting rural development. Xie et al. [24]measured urban–rural integration at the city level using multi-source data. Yang et al [25] explored evaluation methods for high-quality development of agricultural economy. Tang et al. [26] investigated evaluation methods for the level of cultural-tourism integration development in traditional villages.

Rural areas are complex territorial systems with multifaceted structural elements [12]. In the quantitative evaluation of rural development, the establishment of an evaluation index system holds particular significance. Cloke [27] initially introduced the concept of a rurality index in the 1970s. Vicki et al. [28] further elaborated on the rurality index proposed by Cloke, and Woods [29] extended Cloke's rurality index system. Many scholars [19,30,31] improved the research framework for the rurality index, considering research methods, index selection, and weight determination. There is a wealth of research findings on the evaluation of rural development levels. For example, Zhang et al. [32], Qian et al. [33], and Qin et al. [34] established evaluation index systems for agricultural and rural development levels at varying scales. Zhang [22], Liang et al. [35], and Chen et al. [36] evaluated agricultural and rural development levels and explored the associated obstacles at national, regional, and provincial levels. Most studies primarily focus on countries or regions at the research scale, such as Qin et al. [37], Xin et al. [38], and Zhang et al. [39], who measured and ranked rural modernization in China.

Qin et al. [34] and Luo [40] assessed agricultural and rural development levels in Shandong and Wuhan, respectively.

Regarding the composition of the indicator system, Martin K. et al. [41]constructed an evaluation index incorporating aspects of the economy, society, and environment, grounded in the sustainability of agricultural systems. Waldorf [42]conducted county-level rural development evaluation using criteria such as "population size, population density, proportion of urban population, and distance to the nearest city". Rezaei and Karami [43] and Carof et al. [44] evaluated rural development holistically, considering economic, social, and ecological (environmental) dimensions. Binsswanger et al. [45]appraised rural development based on five facets: farmers themselves, government policies, infrastructure development, geographical location, and natural resources. Qian [33] structured the index system around three levels: agriculture, rural areas, and farmers, whereas Qin [34] constructed an index system with four levels: factor output, development support, output benefits, and multifunctional expansion. Guo and Hu [46] attempted to integrate general requirements for rural revitalization strategy, the "five-in-one" overall layout, prioritized agricultural and rural development, and integrated urban–rural development to establish an evaluation system. Some scholars developed a five-dimensional evaluation system for rural development on the basis of China's "Twenty Character" Policy for Rural Revitalization Strategy [39,47]. These studies primarily focus on developing countries or regions, and their findings provide valuable references.

Few scholars have undertaken a comprehensive global evaluation of rural development across various countries, providing an international perspective on the state of diagnosing the level of rural development in developing countries. These studies that comprehensively compare rural development across countries worldwide shed light on the development trajectories of developed nations, exploring China's unique characteristics and existing gaps, and distill lessons Western countries can offer. These insights serve as valuable references for the rural development strategies of developing countries, including China. In light of this gap in the literature, this study, building upon existing research, constructs an evaluation index system for rural development in countries worldwide to quantitatively assess the rural development levels across nations. The study also explores the constraining factors affecting rural development, identifies the primary disparities between typical developing countries and developed nations, synthesizes the experiences of developed countries, and presents tailored recommendations for future rural development in developing countries.

## Data sources and research methods
### Methodology

Research on the comprehensive evaluation of rural development is continuously enriching. The development of agricultural economies, the social life of rural residents, and rural ecological environments represent the primary focus areas of current studies. Agriculture, rural areas, and farmers are the core elements of rural development, with the well-being of the farmers at its heart [48]. Both agricultural and rural development are equally critical, and key strategies include enhancing agricultural productivity and improving living conditions in rural areas. Historical and contemporary perspectives underscore the importance of adopting a broad historical view when addressing issues related to agriculture, rural areas, and farmers. Problems associated with these three aspects—commonly referred to as 'San Nong' issues—are considered an inevitable outcome of the agricultural society's transformation [49]. This phenomenon is evident in the industrial transformation processes of various countries globally. This manuscript constructs an evaluation index system from the perspectives of agriculture, rural areas, and farmers.

The methodology for the comprehensive evaluation of rural development is becoming increasingly systematic and diversified. Typically, this involves the construction of a complete evaluation system, followed by the assignment of weights to various indicators using specific methods, and the application of scientific evaluation models for calculation. This manuscript employs a combination of subjective Analytic Hierarchy Process (AHP) and objective entropy weight method to determine the weights of each indicator. Subsequently, the Topsis method and others are utilized to derive the final evaluation results [50,51].

## Selection of evaluation indicators

This manuscript draws upon existing research findings and is grounded in adherence to fundamental principles such as scientific validity, representativeness, accessibility, and global applicability, and global applicability, and also takes into consideration the availability of global data, this study establishes an evaluation system comprising 27 specific indicators across eight dimensions. These dimensions include agricultural production, output quality and benefits, green sustainability, agricultural support, social development, human settlement environment, quality of life, and cultural education. These indicators are organized around three subsystems: agricultural development, rural development, and farmers' lives. (Table 1). In addition, the selected indicators were subjected to a collinearity diagnosis, with Variance Inflation Factor (VIF) values below 10, confirming the rationality and representativeness of the indicator selection.

## Determination of the index weight by AHP-entropy method

(1) Determination of subjective weights by AHP method

Twenty experts in this field are invited to subjectively assign weights to the evaluation indexes and create a judgment matrix. Subsequently, hierarchical ordering and consistency testing are conducted to derive the subjective weights for various evaluation indicators. The Consistency Ratio (CR) values of the judgment matrices, 0.097 and 0.056 respectively, are both less than 0.1, indicating that they pass the consistency test.

(2) Determination of objective weights by the entropy weight method

The main steps are data standardization, calculation of information entropy, and the determination of weights based on information entropy.

(3) Combined weighting of evaluation indexes

The AHP method is employed to assign weights to the three criteria layers and eight sub-criteria layers of the evaluation index system. Subsequently, the entropy method is utilized to ascertain the weights of the standard layer within the criteria layer. Finally, the combined weights are calculated.

## TOPSIS model

The TOPSIS comprehensive evaluation method assesses research objects by ranking their distance from the evaluation object to the optimal and inferior solutions. This method is widely used in various evaluation studies [51,52].

Step 1. Construct a weighted matrix $Z$.

$$Z = y_{ij} * w_j \tag{1}$$

**Table 1. Evaluation index system of rural comprehensive development level.**

| Criterion layer | Subcrite-rion layer | Standard layer | Description | Data sources | Attrib-utes |
|---|---|---|---|---|---|
| Agricul-tural devel-opment | Agricultural production | Arable land (ha/ person) | Average hectare per person:Arable land (hect-ares)/Population, total | https://data.worldbank.org/indicator/AG.LND.ARBL.HA.PC | + |
| | | Employment in agriculture as % of total employment | Percentage of employed personnel in agricul-ture to total employment | https://data.worldbank.org/indicator/SL.AGR.EMPL.ZS | – |
| | | Agricultural machinery (set) | Number of tractors per 100 km² of cultivated land | https://www.fao.org/faostat/zh/#dataRY | + |
| | | Crop production index | Annual agricultural production in the base period from 2004 to 2001, including all crops except forage crops | https://data.worldbank.org/indicator/AG.PRD.CROP.XD | + |
| | | Livestock production index | Annual animal husbandry production in the base period from 2016 to 2014 | https://data.worldbank.org/indicator/AG.PRD.LVSK.XD | + |
| | Output quality and benefits | Agricultural GDP as % of total GDP | Percentage of agricultural added value to total production value; Agricultural GDP/ GDP | Agricultural GDP: https://data.world-bank.org/indicator/NV.AGR.TOTL.KD GDP: https://data.worldbank.org/indica-tor/NY.GDP.MKTP.CD | – |
| | | Cereal yield (kg per ha) | Kilograms Cereal yield per hectare | https://data.worldbank.org/indicator/AG.YLD.CREL.KG | + |
| | | Agricultural land output rate (US/ha) | Agriculture, forestry, and fishing, value added (current US$)/Cultivated land area | Agriculture, forestry, and fishing, value added (current US$): https://data.worldbank.org/indicator/NV.AGR.TOTL.CD Cultivated land area: https://www.fao.org/faostat/zh/#data/RL | + |
| | Green sustainable | Agricultural water consumption (m³/10,000 US dollars) | Annual freshwater withdrawals, agriculture (% of total freshwater withdrawal)/Agriculture, forestry, and fishing, value added (current US$) | Annual freshwater withdrawals, agriculture: https://data.worldbank.org/indicator/ER.H2O.FWAG.ZS Agriculture GDP (current US$): https://data.worldbank.org/indicator/NV.AGR.TOTL.CD | – |
| | | Pesticide consumption (kg/ha) | Pesticides Use/Cultivated land area | Pesticides Use: https://www.fao.org/faostat/zh/#data/RP Cultivated land area: https://www.fao.org/faostat/zh/#data/RL | – |
| | | Fertilizer consumption(kg/ha) | kilograms per hectare of arable land: Total fertilizer use/Cultivated land area | https://data.worldbank.org/indicator/AG.CON.FERT.ZS | – |
| | Agricultural support | Intensity of government expenditure on agricultural support (%) | Government expenditure on agricultural sup-port/Total expenditures | https://data.worldbank.org/indicator/GC.XPN.TOTL.CN | + |
| | | Intensity of financial expenditure on agri-cultural support (%) | Financial expenditure on agricultural support/Added value of agriculture | https://data.worldbank.org/indicator/GC.XPN.TOTL.CN | + |

*(Continued)*

**Table 1.** (Continued)

| Criterion layer | Subcriterion layer | Standard layer | Description | Data sources | Attributes |
|---|---|---|---|---|---|
| Rural development | Social development | Urbanization rate(%) | Urban population/Total population | https://data.worldbank.org/topic/urban-development | + |
| | | Proportion of rural population with electricity available (KWh/person) | Electricity penetration rate among rural population – | https://unstats.un.org/sdgs/dataportal/database (Data Series 7.1.1) | + |
| | | Proportion of rural population using basic drinking water services (%) | Rural population water penetration rate | https://unstats.un.org/sdgs/dataportal/database (Data Series 1.4.1) | + |
| | | Physicians (per 1,000 people) | Number of physicians per thousand population | https://data.worldbank.org.cn/indicator/SH.MED.PHYS.ZS | + |
| | | Individuals using the Internet (% of population) | Percentage of Individuals using Internet to total population | https://data.worldbank.org.cn/indicator/IT.NET.USER.ZS | + |
| | Human settlement environment | Forest area (% of land area) | Percentage of area of woods to land area | https://data.worldbank.org/indicator/AG.LND.FRST.ZS | + |
| | | Density of road network (person/km²) | Kilometers of highway network/Area of the territory | https://mp.weixin.qq.com/s/vUloBYAM1gcCehn07g4ecA | + |
| Farmers' life | Quality of life | Income per capita (US dollars) | Per capita income of each adult | https://wid.world/zh/world-cn/#anninc_p0p100_z/US;FR;DE;CN;ZA;GB;WO/last/eu/k/p/yearly/a/false/0/75000/curve/false/country | + |
| | | Prevalence of undernourishment (% of population) | Proportion of undernourished individuals to total population | https://data.worldbank.org.cn/indicator/SN.ITK.DEFC.ZS | – |
| | | Life expectancy at birth, total (years) | Represented by life expectancy | https://data.worldbank.org.cn/indicator/SP.DYN.LE00.IN | + |
| | | Gini index | Reflecting wealth disparity among residents | https://data.worldbank.org.cn/indicator/SI.POV.GINI | – |
| | Cultural education | Completion rate of higher education (%) | Proportion of rural population completing high school education | https://unstats.un.org/sdgs/dataportal/database (Data Series 4.1.2) | + |
| | | Average length of education | Average length of education of the adults aged above 25 in various countries | https://hdr.undp.org/data-center/documentation-and-downloads (DATA LINKS Table 1 SDG4.4) | + |
| | | Government expenditure on education, total (% of GDP) | Percentage of total public expenditure on education in the total expenditure | https://data.worldbank.org.cn/indicator/SE.XPD.TOTL.GD.ZS | + |

Step 2. Determine the optimal solution $Z_j^+$ and the inferior solution by basing on the weighted matrix constructed in the previous step $Z_j^-$.

$$Z_j^+ = \left\{ maxZ_{ij} | i = 1,2,\ldots n \right\} \tag{2}$$

$$Z_j^- = \left\{ minZ_{ij} | i = 1,2,\ldots n \right\} \tag{3}$$

Step 3. Calculate the Euclidean distance D from various index schemes to the optimal solution $Z_j^+$ and the inferior solution $Z_j^-$.

$$D_i^+ = \sqrt{\sum_{j=1}^{m} \left( Z_j^+ - Z_{ij} \right)^2}, \ i = 1,2,\ldots n \tag{4}$$

$$D_i^- = \sqrt{\sum_{j=1}^{m} \left( Z_j^- - Z_{ij} \right)^2}, \ i = 1,2,\ldots n \tag{5}$$

Step 4. Evaluate the rural comprehensive development level.

$$E_i = \frac{D_i^-}{D_i^+ + D_i^-} \tag{6}$$

Where $E_i$ refers to comprehensive rural development level of the $i^{th}$ country. A larger value indicates a higher comprehensive rural development level.

## Obstacle model

To further analyze the factors constraining the comprehensive development level of rural areas in various countries, this study determines obstacle factors affecting comprehensive development progress. This analysis is based on the comprehensive evaluation results of rural development level, utilizing factor contribution degree, deviation degree, and obstacle degree of indexes, combined with the obstacle model analysis method.

Factor contribution degree ($u_j$) refers to the influence of individual indexes on overall objective and is represented by index weight (Eq 7). Index deviation degree ($I_{ij}$) refers to the distance between single index and development objective, that is, the distance from standardized value of index to 100% (Eq 8). Obstacle degree ($O_{ij}$) quantifies the hindrance level of each index to the comprehensive development level of rural areas and is calculated using Eq 9:

$$u_j = w_j \tag{7}$$

$$I_{ij} = 1 - y_{ij} \tag{8}$$

$$O_{ij} = \frac{I_{ij}u_j}{\sum_{j=1}^{n} I_{ij}u_j} \tag{9}$$

## Influencing factors

In this study, multiple linear regression model [53] is used to establish relationships between the comprehensive development level in rural areas (dependent variable) and multiple factors $X_1, X_2, \ldots, X_n$ (independent variables).

Rural areas are complex regions influenced by multiple factors, and comprehensive rural development is likewise the result of multiple factors. Economic, industrial, geographical, demographic, resource-based, and topographic all impact development in rural areas. Basing on existing research [54–56], we select measurement indexes related to natural resources, economic and industrial development, population and social development, and transportation infrastructure as the main influencing factors. The selection is made with consideration of the representativeness and accessibility of these indexes. Specifically, it includes:

(1)  Background factors of natural resources: Proportion of low-altitude land area, average precipitation depth, per capita cultivated land area, and annual average proportion of the population affected by natural disasters.

(2)  Economic and industrial development: Per capita GDP, the proportion of the value of imports and exports to GDP, and government expenditure on agricultural support.

(3)  Population development: Urbanization ratio, population density, proportion of labor force, and proportion of completion rate of higher education.

(4)  Transportation infrastructure: Road density, railway density, cargo terminal throughput, and air transportation volume.

## Data sources and processing

The data primarily originates from the World Bank (https://data.worldbank.org.cn/) and the Food and Agricultural Organization of the United Nations (https://www.fao.org/home/zh/). It includes data on agricultural production, agricultural output quality, government expenditure on agricultural support, rural social development, natural environment, and socio-economic development. The data is consistently standardized, encompassing basic information from the majority of countries globally, and is highly reliable for global-level comparisons. It is primarily based on 2020 data, with any missing 2020 data replaced by the latest available data. Countries with significant missing data are excluded. The dataset comprises a total of 175 countries and regions. The World Bank's public database, which aggregates data at the national level, primarily sources its data from the statistical systems of its member countries. These systems adhere to internationally recognized standards and norms, thereby ensuring reliable and high-quality data. The Bank's various regions and departments collaborate closely to maintain stringent quality control and integrity of the data throughout its collection, compilation, and dissemination. This data is published in various formats in an extensive array of data publications. The FAO Statistical Database, managed by the United Nations, is the largest global database on food, agriculture, and natural resource management. Verified and published by subject experts and statistical personnel, this database provides high-quality data to assist governments in enhancing their operational quality. These data, having consistent statistical standards and covering the basic information of the vast majority of countries worldwide, are highly credible and suitable for global comparisons.

The objective of this manuscript is to assess the current state of rural development across a range of countries globally. Consequently, the manuscript primarily employs the most comprehensive and recent data available from the year 2020. In instances where certain indicators were absent, interpolation was employed using the most recent available data. Furthermore, countries and regions with substantial data gaps, such as Gibraltar and Monaco, were excluded from the analysis. Similarly, regions with low comparability, such as the Channel Islands and the Cayman Islands, were also excluded from the analysis. In conclusion, a total of 175 countries and regions were included in the evaluation.

## Results and analysis

### Evaluation of rural comprehensive development level in various countries worldwide

**General conditions.** The combination weighing process involves the adoption of the AHP and entropy methods. The consistency ratios (CR) obtained from the CR test judgment matrices are 0.097 and 0.056, both falling below the 0.1 threshold, thus satisfying the consistency test. The final weightings are displayed in Table 2. At the criterion level, the weights assigned to agricultural development, rural development, and farmers' living standards are 0.569, 0.242, and 0.189, respectively.

**Table 2. Combined weight table for the evaluation index system of rural comprehensive development level.**

| Criterion layer | AHP weight | Subcriterion layer | AHP weight | Standard layer | Entropy weight | Combination weight |
|---|---|---|---|---|---|---|
| **Agricultural development** | 0.569 | Agricultural production level | 0.384 | Arable land (ha/ person) | 0.276 | 0.060 |
| | | | | Employment in agriculture as % of total employment | 0.058 | 0.013 |
| | | | | Agricultural machinery (set) | 0.538 | 0.117 |
| | | | | Crop production index | 0.089 | 0.019 |
| | | | | Livestock production index | 0.040 | 0.009 |
| | | Output quality and benefits | 0.324 | Agricultural GDP as % of total GDP | 0.035 | 0.006 |
| | | | | Cereal yield (kg per ha) | 0.253 | 0.047 |
| | | | | Agricultural land output rate (US dollars/ha) | 0.712 | 0.131 |
| | | Green sustainable | 0.17 | Agricultural water consumption (m³/10,000 US dollars) | 0.477 | 0.046 |
| | | | | Pesticide consumption (kg/ha) | 0.278 | 0.027 |
| | | | | Fertilizer consumption(kg/ha) | 0.245 | 0.024 |
| | | Agricultural support intensity | 0.122 | Government expenditure on agricultural support (%) | 0.735 | 0.051 |
| | | | | Intensity of financial expenditure on agricultural support (%) | 0.265 | 0.018 |
| **Rural development** | 0.242 | Social development level | 0.725 | Urbanization rate (%) | 0.165 | 0.029 |
| | | | | Proportion of rural population with electricity available (KWh/person) | 0.135 | 0.024 |
| | | | | Proportion of rural population using basic drinking water services (%) | 0.104 | 0.018 |
| | | | | Physicians (per 1,000 people) | 0.451 | 0.079 |
| | | | | Individuals using the Internet (% of population) | 0.146 | 0.026 |
| | | Human settlement environment | 0.275 | Forest area (% of land area) | 0.417 | 0.028 |
| | | | | Density of road network (person/km²) | 0.583 | 0.039 |
| **Farmers' life** | 0.189 | Quality of life | 0.725 | Income per capita (US dollars) | 0.674 | 0.092 |
| | | | | Prevalence of undernourishment (% of population | 0.079 | 0.011 |
| | | | | Life expectancy at birth, total (years) | 0.163 | 0.022 |
| | | | | Gini index | 0.085 | 0.012 |
| | | Cultural education level | 0.275 | Completion rate of education (%) | 0.426 | 0.022 |
| | | | | Average length of education | 0.311 | 0.016 |
| | | | | Government expenditure on education, total (% of GDP) | 0.263 | 0.014 |

The overall evaluation results for rural comprehensive development levels are illustrated in Fig 1, with an average score of 0.382. The natural breakpoint method is employed to categorize comprehensive rural development into five intervals: high level (above 0.462), moderate to high level (0.413–0.462), moderate level (0.368–0.413), moderate to low level (0.324–0.368), and low level (below 0.324) (Fig 2). Notable countries with the highest rural development levels globally include Switzerland, Japan, South Korea, Germany, and the United States. Countries with the lowest rural development levels consist of Afghanistan, Myanmar, Pakistan, Kenya, South Sudan, and other less developed nations. China, as the world's largest developing country, stands at a moderate level of rural comprehensive development with a score of 0.39, slightly surpassing the global average but still trailing behind developed nations.

From a spatial perspective, rural areas in North America, Oceania, and Europe exhibit relatively high development levels, exemplified by countries like Canada, Australia, and Germany. In contrast, most rural areas in Africa fall below the global average level, including nations such as Egypt, Kenya, and the Congo. Asia and South America generally fare well, mostly achieving moderate or higher levels of development.Regions such as North America, which benefit from higher natural, economic, and social levels, exhibit correspondingly advanced levels of rural comprehensive development. Conversely, regions such as Africa, where economic development is significantly lagging behind that of Europe and other regions, exhibit comparatively poorer levels of rural development. This pattern demonstrates a clear dichotomy between the northern and southern regions, with the former exhibiting higher levels of development than the latter.

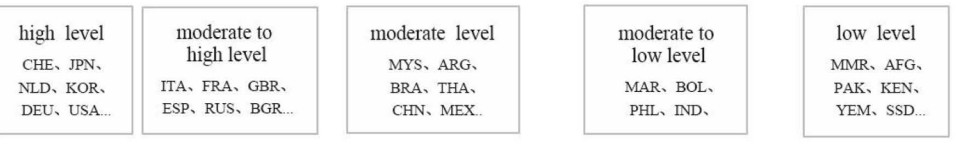

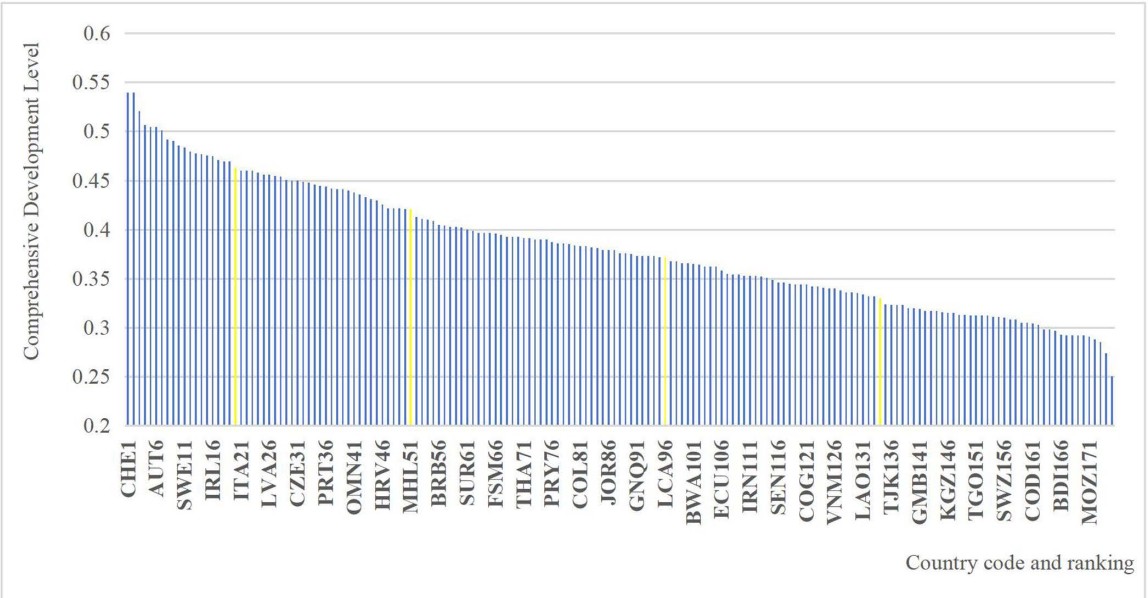

**Fig 1. Global ranking of countries based on rural comprehensive development level.**

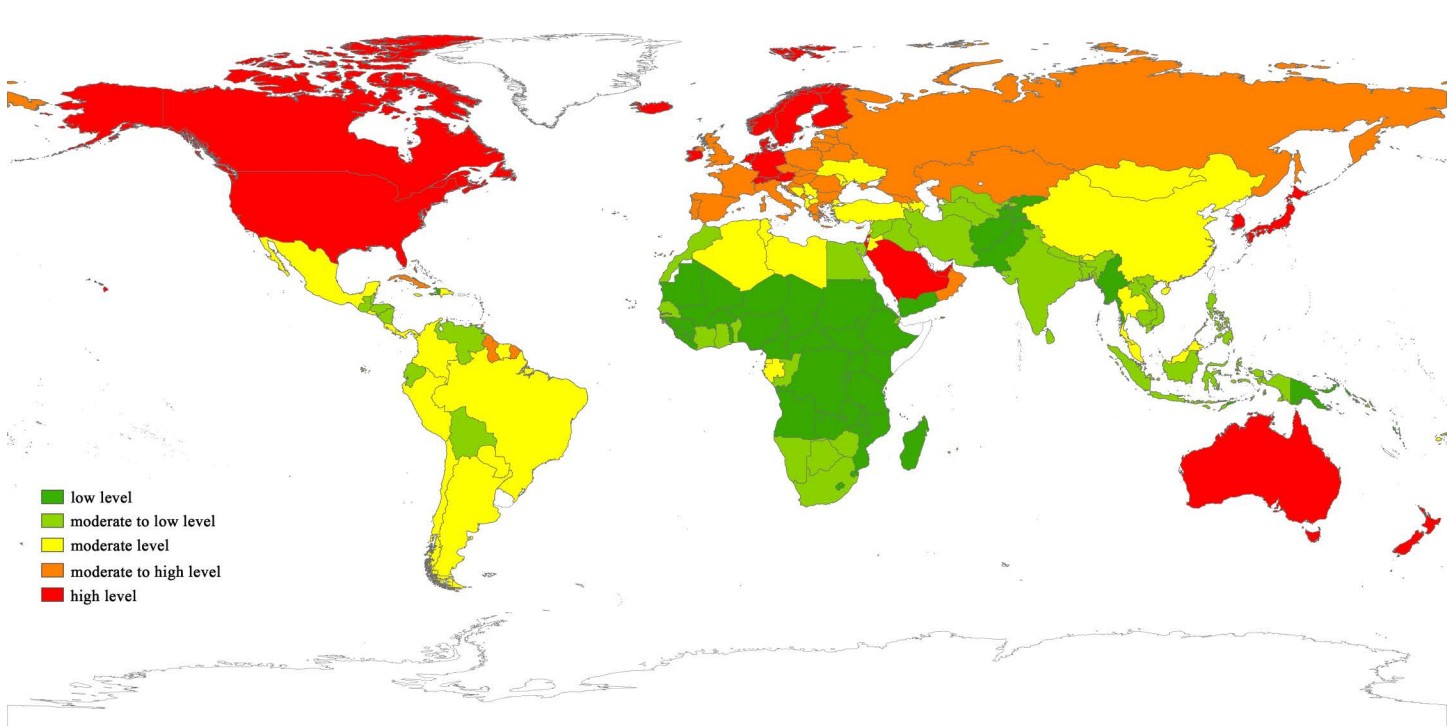

**Fig 2. World map depicting the distribution of rural comprehensive development level.**

## Three-dimensional diagram of agriculture, rural areas, and farmers

In this study, by applying a three-dimensional K-means clustering analysis, we identified distinct patterns of performance across the key dimensions of agriculture, rural areas, and farmers among various countries. Three prominent clusters were described, representing different performance patterns among national groups(Fig 3):

(1) First Cluster (Red): This cluster encompasses countries that exhibit a relatively balanced and high performance across the indicators of agricultural development, rural development, and farmer development, such as Japan, Austria, and Norway. These nations demonstrate a strong coupling between agricultural progress and farmer welfare, reflecting a high level of comprehensive agricultural development.

(2) Second Cluster (Green): Includes countries with high levels of agricultural development but lagging farmer and rural development, such as the United Arab Emirates, Switzerland, and the Netherlands. The imbalance in this development pattern may indicate that despite advanced agricultural technology and output, overall farmer welfare and the progression of rural infrastructure have not kept pace.

(3) Third Cluster (Blue): Covers countries that perform poorly across all three development indicators, such as Brazil, Maldives, and Indonesia. These nations face common challenges in agriculture and rural development and urgently require interventions through domestic policies and international support to enhance their overall development levels.

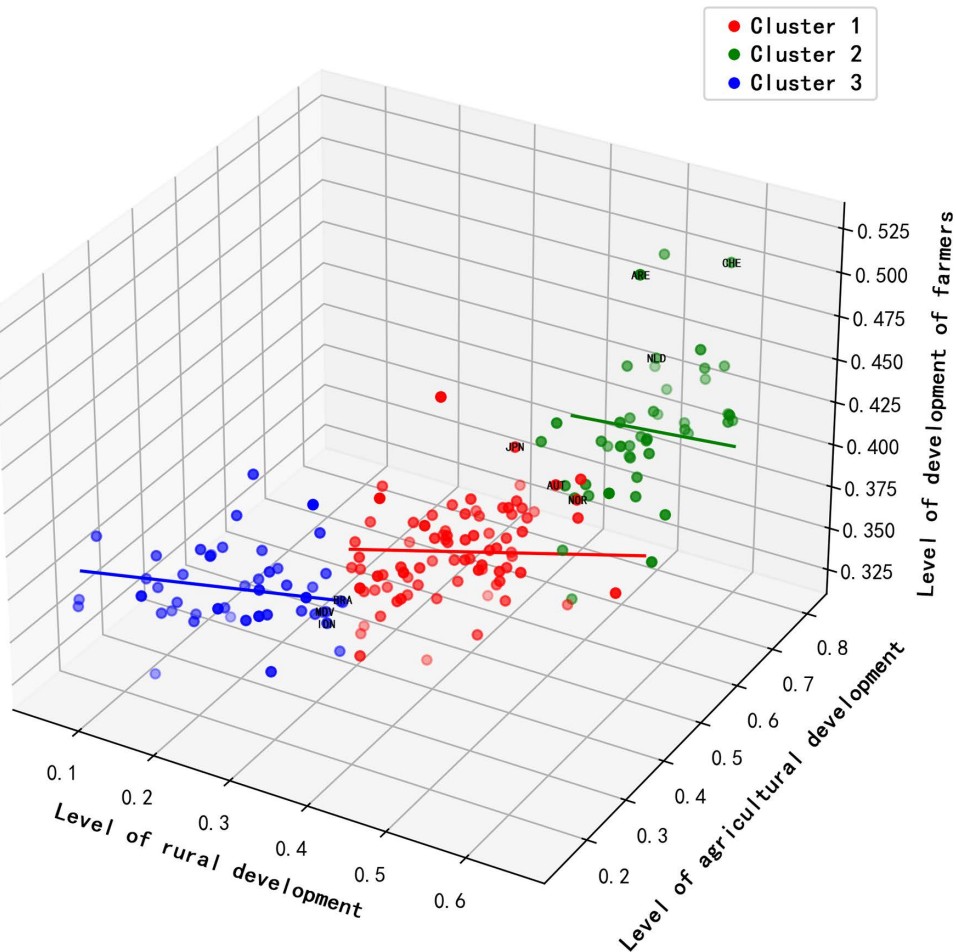

**Fig 3. Map of three-dimensional spatial patterns of countries of the world.**

**Classification and comparative analysis.**

(1)  Income type and rural comprehensive development level

The World Bank classifies world economies into four categories based on per capita gross national income: low income (<$1035), medium to low income ($1036–$4045), medium to high income ($4046–$12535), and high income (>$12536). A significant disparity in rural development levels is observed among different income types (Fig 4), with high-income countries showcasing substantially superior rural development compared with the other three categories.

(2)  Per capita GDP and rural comprehensive development level

Countries are categorized into four levels based on per capita GDP: high poverty level (<$2000), poverty level ($2000–$5000), moderate level ($5000–$10896), and developed level (>$10896, i.e., the world average in 2020). Fig 5 demonstrates a close relationship between rural development levels and economic status. However, significant variations persist among countries at the same development level. Some highly developed countries, such as Argentina, Turkey, Panama, and Chile, exhibit relatively lower rural development levels.

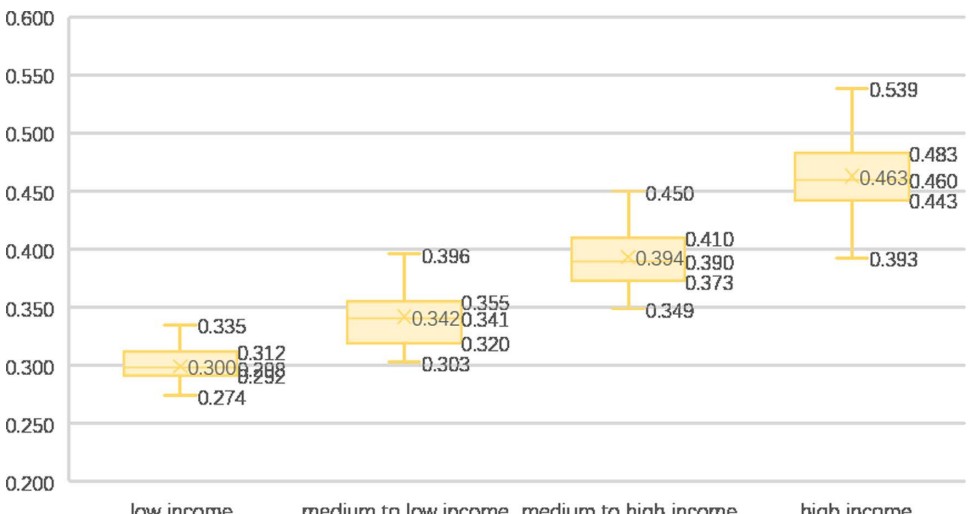

**Fig 4. Rural comprehensive development in countries with different income types.**

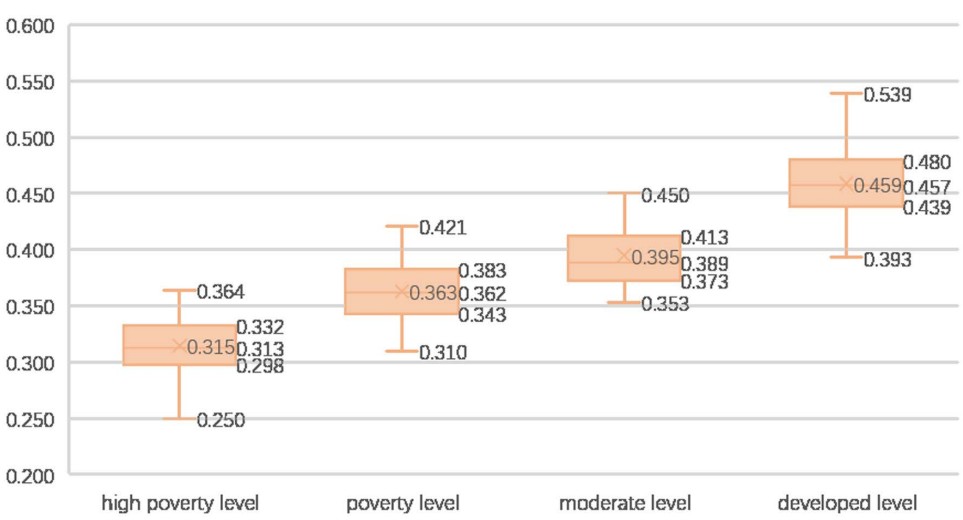

**Fig 5. Rural comprehensive development in countries with different per capita GDP levels.**

(3) HDI and rural comprehensive development level

Countries are divided into four categories based on their HDI scores in 2020: low HDI (<0.550), moderate HDI (0.550–0.699), high HDI (0.700–0.799), and extremely high HDI (≥0.800). Fig 6 illustrates that the rural comprehensive development level is closely aligned with HDI. Nations with high and extremely high HDI scores boast high or relatively high rural comprehensive development levels. Among low HDI countries, 25% exhibit moderate to low levels of rural development, whereas 75% are characterized by relatively lower development levels.

(4) Urbanization level and rural comprehensive development level

Countries are classified into three stages of urbanization on the basis of the 30% and 70% thresholds: initial stage, accelerated stage, and later stage. A clear intersection between

urbanization level and rural comprehensive development is discernible, as illustrated in Fig 7. During the initial and accelerated stages of urbanization, countries tend to exhibit lower rural comprehensive development levels, whereas the late stage of urbanization generally corresponds to higher development levels.

(5)  Spearman rank correlation coefficient analysis

Table 3 demonstrates the correlation between the rural comprehensive development level and various factors such as income type, per capita GDP, HDI, and urbanization rate. The ranking of rural comprehensive development level exhibits strong correlations with income type, per capita GDP, and HDI, as indicated by correlation coefficients of 0.935, 0.934, and 0.937, respectively. However, the correlation between the ranking of urbanization rate and rural comprehensive development level is relatively modest at 0.693. This suggests that the ranking of rural comprehensive development aligns with overall national income, per capita GDP, and HDI ranking but diverges significantly from the ranking of urbanization levels. For instance, Panama, despite being a high-income country, maintains a moderate level of rural development. By contrast, Guyana, still in the initial stage of urbanization, exhibits a moderate to high level of rural development.

## Analysis of obstacle degree to rural comprehensive development

**Obstacle factors.**  To further analyze the key factors hindering the comprehensive development of rural areas in various countries, the obstacle model measures the obstacle degree of individual indexes across countries and identifies the primary obstacle factors that constrain rural comprehensive development. Factors with obstacle degrees ($O_{ij}$) greater than 5% are considered main obstacle factors [32,57], and their frequency of occurrence is summarized.

The eight main obstacle factors affecting rural comprehensive development worldwide are agricultural mechanization level, agricultural land output rate, per unit land area grain yield, arable land per person, government expenditure on agricultural support, physician number, road network density, and per capita income. Additionally, agricultural water consumption and forest area proportion are identified as obstacle factors in some countries, but less frequently.

## Differences in obstacle factors between developed and underdeveloped countries

Comparing obstacle factors between developed and underdeveloped countries reveals notable differences. Common obstacle factors across both groups include agricultural mechanization level, agricultural land output rate, per unit land area grain yield, the physicians number and arable land per person. In underdeveloped countries, per capita income is a prominent obstacle factor. Conversely, developed countries face challenges related to government expenditure on agriculture, among others.

## Factors influencing rural comprehensive development level

The multiple linear regression analysis model yields an adjusted $R^2$ of 0.898 and an F-value of 103.124, both significant at the 0.01 level. Tolerance values exceed 0.1, and the variance inflation factor (VIF) remains below 3 after testing, indicating no significant multicollinearity among independent variables (Table 4). The Durbin–Watson value stands at 2.084, suggesting that residuals are independent, and standardized residuals approximate a normal distribution. These results affirm the reasonable setup of the model.

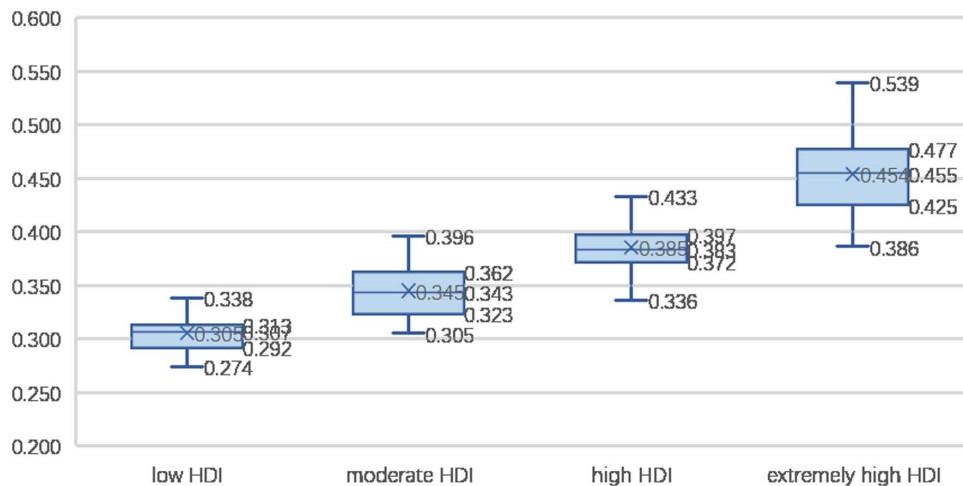

**Fig 6. Rural comprehensive development in countries at different HDI levels.**

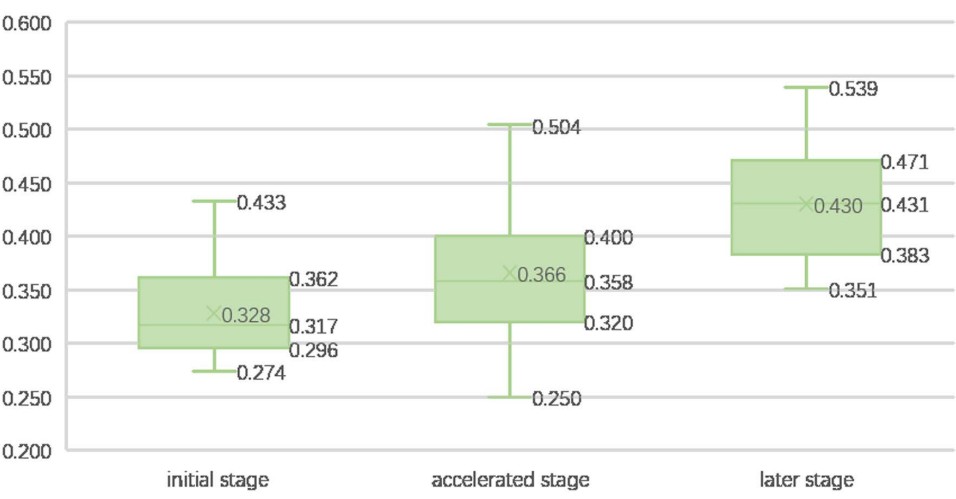

**Fig 7. Rural comprehensive development in countries at different urbanization levels.**

**Table 3. Correlation between rural comprehensive development level and income type, per capita GDP, HDI, and urbanization rate.**

| Correlation | | | Ranking of income types | Ranking of per capita GDP | Ranking of HDI | Ranking of urbanization rate |
|---|---|---|---|---|---|---|
| Spearman Rho | Ranking of comprehensive development level | Correlation coefficient | 0.935 | 0.934 | 0.937 | 0.693 |
| | | sig. | 0.000 | 0.000 | 0.000 | 0.000 |

(1) The proportion of low-altitude land area and per capita cultivated land area in natural resources are the main factors influencing rural comprehensive development. However, average precipitation depth and the annual average proportion of the population affected by natural disasters do not exhibit statistical significance with a p-value greater than 0.05. Topographic conditions have a significant impact on rural industries, transportation, and infrastructure construction. Insufficient per capita cultivated land is a crucial factor restricting rural development in numerous countries.

(2) Concerning economic and industrial development, per capita GDP, the proportion of the value of imports and exports to GDP, and government expenditure on agricultural support are significant factors influencing rural comprehensive development. Per capita GDP has a highly significant impact on rural comprehensive development, as indicated by a standardized regression coefficient of 0.319.

(3) In terms of population development, factors such as urbanization ratio, population density, and the completion rate of higher education influence rural comprehensive development. The completion rate of higher education has the highest standardized coefficient among all influencing factors, at 0.529. However, the proportion of the labor force does not pass the significance test of $p < 0.05$, suggesting that the relationship between population structure and the level of rural comprehensive development is not statistically significant.

(4) Regarding transportation infrastructure, the density of the road network influences the comprehensive development level of rural areas. However, factors such as railroad density, cargo terminal throughput, and air transportation volume exhibit negligible impacts on rural comprehensive development.

## Discussion

(1) In general, the level of rural comprehensive development is highly correlated with national income level, per capita GDP, and HDI level. However, the correlation between the level of rural comprehensive development and urbanization level is relatively weak. During the accelerated stage of urbanization, urbanization levels advance rapidly, and the comprehensive level of rural areas often significantly improves only in the later stages of urbanization. In the initial stage of urbanization, agriculture still predominates. Neglecting rural areas while prioritizing urban areas can result in severe urban–rural imbalances and a decline in rural areas during urbanization In the acceleration stage of urbanization [58].

(2) The results reveal three main categories of obstacle factors to rural comprehensive development: agricultural development (including agricultural mechanization level, agricultural land output rate, per land area of grain yield, average per labor of cultivated land area, and government expenditure on agricultural support), rural development (including the physicians number and the density of the road network), and farmers' life (per capita income).

On a global scale, indicators related to agricultural production, such as green sustainable development level and crop production index, have relatively low obstacle degrees to rural development. Although countries vary significantly in green sustainable development levels, the obstacle degree remains relatively low. Crop production index and livestock production index show little variation among different countries. Improving agricultural

**Table 4. Fitting result of regression model.**

| Independent variable | Standardized Coefficient | t | Significance | Tolerance | VIF |
|---|---|---|---|---|---|
| Proportion of low-altitude land area | 0.090 | 2.937 | 0.004 | 0.628 | 1.592 |
| Average precipitation depth (mm per year) | 0.024 | 0.851 | 0.396 | 0.732 | 1.365 |
| Per capita cultivated land area | 0.100 | 3.548 | 0.001 | 0.743 | 1.347 |
| Annual average proportion of population affected by natural disasters | -0.047 | -1.659 | 0.099 | 0.721 | 1.386 |
| Per capita GDP | 0.319 | 8.248 | 0.000 | 0.393 | 2.547 |
| Proportion of import and export value to GDP | 0.092 | 3.542 | 0.001 | 0.869 | 1.150 |
| Government expenditure on agricultural support | 0.092 | 3.124 | 0.002 | 0.671 | 1.490 |
| Urbanization rate | 0.188 | 5.408 | 0.000 | 0.484 | 2.065 |
| Population density | 0.062 | 2.124 | 0.035 | 0.682 | 1.465 |
| Proportion of labor force | -0.031 | -1.042 | 0.299 | 0.657 | 1.521 |
| Completion rate of higher education | 0.529 | 13.183 | 0.000 | 0.365 | 2.739 |
| Road density | 0.077 | 2.544 | 0.012 | 0.636 | 1.571 |
| Railway density | -0.006 | -0.219 | 0.827 | 0.751 | 1.331 |
| Cargo terminal throughput | 0.060 | 1.722 | 0.087 | 0.481 | 2.079 |
| Air transportation volume | -0.052 | -1.496 | 0.137 | 0.477 | 2.098 |
| (Constant) | | 19.445 | 0.000 | | |

production efficiency and increasing government expenditure on agricultural support remain the primary pathways for the vast majority of countries to enhance the comprehensive development level of rural areas. In terms of rural development, factors such as electricity, drinking water, networks, and other infrastructure, along with the Forest area proportion in rural areas, are not core obstacles. However, global disparities in healthcare levels and road network density remain significant. In terms of farmers' life, this study employs numerous indicators for comprehensive evaluation, but only per capita income serves as an obstacle factor. This underscores that enhancing per capita income remains a central lever for improving rural living standards. Per capita income directly measures farmers' life [59], which is the ultimate goal of rural revitalization and the most pressing obstacle to address

(3) Regarding influencing factors, precipitation and natural disasters no longer pose significant obstacles to modern agriculture development. Precipitation plays a crucial role in agriculture as the primary source of soil moisture and crop water demand [60]. However, recent worldwide investments in farmland water conservancy facilities have significantly improved agricultural infrastructure, enhancing the ability to agricultural disasters prevention capacity. Consequently, the impact of natural disasters on rural comprehensive development is limited, with a noticeable decline in the rate of crops affected by disasters in recent years [61]. The proportion of the labor force can indirectly influence rural productivity levels. Scholars such as Wu et al. [62] and Li et al. [63] believed that substantial labor outflows from rural areas can affect agricultural production and resource allocation efficiency to some extent, potentially diminishing the sustainable development capacity of rural areas. However, in the era of agricultural mechanization, the labor force's impact on rural development has significantly diminished, giving way to the education level of the working population. The completion rate of higher education emerges as a pivotal factor influencing rural development. As education levels improve, rural talent, technology, organization, and various other aspects can see substantial enhancement, thereby promoting an overall improvement in rural comprehensive development level [64].

## Conclusions and suggestions

### Conclusions

Drawing on prior research, this study has devised an evaluation index framework for rural development encompassing agricultural, rural, and farmer dimensions. It aims to comprehensively assess and juxtapose rural development across diverse countries. Both obstacle and regression models are employed to pinpoint hindrances and influencing factors. Additionally, this research has distilled the experiences of developed nations, formulating specific recommendations for China's future rural development. The results indicate the following:

(1) Significant disparities in the comprehensive development of rural areas exist on a global scale. Moreover, the rural comprehensive development exhibits both connections and distinctions from national income level, per capita GDP, HDI, and urbanization level. Notably, rural comprehensive development demonstrates a weaker correlation with urbanization level compared with the other three indicators.

(2) The three main categories of obstacle factors to rural comprehensive development are agricultural development (agricultural mechanization, agricultural land output rate, per land area of grain yield, agricultural GDP, and government expenditure on agricultural support), rural development (the physician number and density of road network), and farmers' life (per capita income). To advance rural comprehensive development, most countries should prioritize improving agricultural production efficiency, increasing government expenditure on agricultural support, and enhancing per capita income.

(3) Factors influencing rural comprehensive development levels in various countries encompass the proportion of low-altitude land area, per capita cultivated land area, per capita GDP, proportion of imports and exports to GDP, government expenditure on agricultural support, urbanization rate, completion rate of higher education, and road network density. Notably, precipitation and natural disasters, population density, and the proportion of the labor force no longer serve as significant obstacles to modern agriculture development. Instead, the education level of the working population emerges as the most pronounced impact of rural development.

### Suggestions

Building upon these research findings and considering the particular circumstances of developing nations, including China, the following recommendations are put forward to improve rural development in the future:

(1) In terms of agricultural development, to enhance agricultural labor productivity, augment government expenditure on agricultural support, and increase capital investment in agricultural mechanization. In terms of cultivated land use, especially under limited per capita cultivated land area, it is advisable to raise the agricultural mechanization rate, improve arable land per person, and enhance output benefits. Concurrently, the construction of high-standard agricultural land should be actively pursued, promoting agricultural land output rates and elevating average land yields. Structural optimization and augmented financial backing for agriculture should also be prioritized.

(2) In terms of rural development, to enhance the hardware and software facets of rural infrastructure, including healthcare and road networks, thereby enhancing living conditions in rural areas. Increased investments in hospitals and rural medical facilities are essential, alongside an increment in physicians (per 1,000 people) and intensified professional

training for medical personnel. Road network expansion in rural regions is crucial, offering support to rural industries and the livelihoods of rural residents.

(3) In terms of farmers' life, to focus on elevating farmers' income levels, proactively boosting their education levels, and providing comprehensive production and life assurances. Recognizing that farmer income forms the bedrock of happiness and rural development, investments in rural education and talent cultivation should be increased. Attracting talent to initiate rural enterprises, enhancing farmers' educational attainments, driving rural industrial development, and expanding income-generating avenues for farmers are imperative strategies for augmenting their livelihoods.

## Limitations and Future Prospects

[1] The scope of this manuscript is quite broad. Due to the differences in actual development status and historical background among various countries, ensuring consistent and comparable rural evaluation indicators across countries, while comprehensively considering the impact of existing statistical data on a global scale, presents certain challenges in indicator selection. This results in difficulties in comprehensively summarizing indicator factors, leading to the omission of certain indicators and insufficient consideration of local characteristics of different countries. This is an area that needs further improvement in future research, aiming to develop more tailored evaluation indicators for different types of rural areas and to discuss them in a categorized manner.

[2] Rural development is dynamic. Due to difficulties in collecting basic data and thoroughly investigating historical factors, this study focuses only on the current situation, lacking exploration in the time dimension and dynamic evolution. As a result, it cannot deeply investigate the underlying reasons for the development of rural regional systems. This is a key area for future research, and the author will continue to engage in ongoing reflection and exploration in subsequent studies.

## Author contributions

**Writing – original draft:** Yiyong Chen, Ling Zhu.

**Writing – review & editing:** Yiyong Chen, Jinzhao Du, Wuyang Hong.

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
