## [Decision Letter · Decision Letter 0]

1 May 2024

PONE-D-24-07831Evaluation on Rural Comprehensive Development Level and Obstacle Factors in Various Countries around the WorldPLOS ONE

Dear Dr. Hong,

Thank you for submitting your manuscript to PLOS ONE. After careful consideration, we feel that it has merit but does not fully meet PLOS ONE’s publication criteria as it currently stands. Therefore, we invite you to submit a revised version of the manuscript that addresses the points raised during the review process.

We look forward to receiving your revised manuscript.

Kind regards,

Changjian Wang

Academic Editor

PLOS ONE

Journal Requirements:

"Fund� Major Program of the National Natural Science Foundation of China�42293273�."

3. Thank you for uploading your study's underlying data set. Unfortunately, the repository you have noted in your Data Availability statement does not qualify as an acceptable data repository according to PLOS's standards.

5. Please ensure that you include a title page within your main document. You should list all authors and all affiliations as per our author instructions and clearly indicate the corresponding author.

6. We note that [Figure 2] in your submission contain [map/satellite] images which may be copyrighted. All PLOS content is published under the Creative Commons Attribution License (CC BY 4.0), which means that the manuscript, images, and Supporting Information files will be freely available online, and any third party is permitted to access, download, copy, distribute, and use these materials in any way, even commercially, with proper attribution. For these reasons, we cannot publish previously copyrighted maps or satellite images created using proprietary data, such as Google software (Google Maps, Street View, and Earth). For more information, see our copyright guidelines: http://journals.plos.org/plosone/s/licenses-and-copyright.

Reviewers' comments:

Reviewer's Responses to Questions

**Comments to the Author**

1. Is the manuscript technically sound, and do the data support the conclusions?

Reviewer #1: Yes

Reviewer #2: Yes

2. Has the statistical analysis been performed appropriately and rigorously? 

Reviewer #1: Yes

Reviewer #2: No

3. Have the authors made all data underlying the findings in their manuscript fully available?

Reviewer #1: Yes

Reviewer #2: No

4. Is the manuscript presented in an intelligible fashion and written in standard English?

Reviewer #1: Yes

Reviewer #2: Yes

5. Review Comments to the Author

Reviewer #1: The manuscript takes a global perspective and evaluates the rural comprehensive development level of 175 countries and regions using a 27-index indicator system, covering agricultural development, rural development, and farmers’ lives. The study uses AHP-entropy method, TOPSIS, obstacle model, and regression model to analyze the rural development level and obstacles. It provides valuable references for developing countries to implement rural revitalization strategies and promotes rural regional system transformation. It also puts forward strategic recommendations to improve rural development, which has certain application value. Here are some suggestions for the improvement of this manuscript.

1. Regarding the selection of indicators, please elaborate on the construction logic of the indicator system and the comprehensiveness and representativeness of the selected indicators to ensure the scientific evaluation.

2. For the data, please provide detailed information on the data processing process, such as data quality assessment, dimension conversion, and handling of outliers, to improve the scientificity of the data.

3. Out of more than 200 countries and regions in the world, why were only 175 (80%) countries and regions evaluated? Which countries were excluded?

4. For exploring the factors influencing rural development level, the linear regression model used is relatively simple and has limitations in handling complex relationships. Consider using more advanced nonlinear regression analysis methods like structural equation models to better reveal the internal relationship between rural development level and influencing factors. Also, why were the 15 indicators in Table 5 selected? Are there any redundant indicators?

Additionally, there are some other minor issues and suggestions.

1. The article title “Evaluation on Rural Comprehensive Development Level” should be changed to “Evaluation of Rural Comprehensive Development Level”.

2. Figure 1 contains Chinese characters.

3. The color differentiation in Figure 2 is too bad.

4. Considering the international audience of the journal, it is suggested to increase the citation of English literature and reduce the citation of Chinese literature.

5. The citation format of some literature is not standardized, such as “Malhoit, Gregory C., 2005” should be changed to “Malhoit, G.C., 2005”. The citation “Research Group of the Rural Economy Research Department” is incorrect.

Reviewer #2: 1、 The abstract appears to have missing parts and the research results are not represented

2、 The introduction seems to be thorough. No comments.

3、 I suggest to add a theoretical framework on the level of integrated rural development to better reflect the theoretical value.

4、 The title data sources and research methods, It seems to have been disordered, suggesting that the methodology be brought forward

5、 Is AHP validated? What are the results of the consistency validation?

6、 How is the comprehensive weight of the indicator system calculated? (Table 3) What are the weights of the two parts (AHP, entropy weight)? Need to be reflected in the article.

7、 Please provide more details regarding the input data used throughout this research. Some supporting references would be helpful for the future readers to justify the data selection.

8、 At the results and analysis section, i suggest to draw figures of spatial pattern in three dimensions�agricultural development, rural development, farmers’ lives�, in addition to the Figure 2.

9、 At the same time, the paper needs to focus on analyzing differences in spatial patterns about 175 countries and regions.

10、 The conclusions section should expand on limitations of this study and future research needs. I suggest listing the bullet points.

11、 The recommendations section could be written separately.

6. PLOS authors have the option to publish the peer review history of their article (what does this mean?). If published, this will include your full peer review and any attached files.

Reviewer #1: No

Reviewer #2: No

---

## [Author Response · Author response to Decision Letter 1]

20 Nov 2024

please check the file Modification Instructions.docx

---

## [Decision Letter · Decision Letter 1]

26 Dec 2024

Evaluation on Rural Comprehensive Development Level and Obstacle Factors in Various Countries around the World

PONE-D-24-07831R1

Dear Dr. Hong,

We’re pleased to inform you that your manuscript has been judged scientifically suitable for publication and will be formally accepted for publication once it meets all outstanding technical requirements.

Kind regards,

Changjian Wang

Academic Editor

PLOS ONE

Review Comments to the Author

Reviewer #1: (No Response)

Reviewer #2: The authors have made revisions to the article to meet the requirements for publication.Some formats still need to be improved according to the requirements of the journal.

---

## [Editor Report · Acceptance letter]

PONE-D-24-07831R1

PLOS ONE

Dear Dr. Hong,

I'm pleased to inform you that your manuscript has been deemed suitable for publication in PLOS ONE. Congratulations! Your manuscript is now being handed over to our production team.

Kind regards,

on behalf of

Prof. Dr. Changjian Wang

Academic Editor

PLOS ONE